# Generational Health Impact of PCOS on Women and Their Children

**DOI:** 10.3390/medsci7030049

**Published:** 2019-03-18

**Authors:** Roger Hart

**Affiliations:** 1Division of Obstetrics and Gynaecology, Medical School, University of Western Australia, Perth, WA 6008, Australia; roger.hart@uwa.edu.au; Tel.: +618 93401322; 2Fertility Specialists of Western Australia, Bethesda Hospital, 25 Queenslea Drive, Claremont, WA 6010, Australia; 3Division of Obstetrics & Gynaecology, King Edward Memorial Hospital, Subiaco, Perth, WA 6008, Australia

**Keywords:** PCOS, offspring, children, pregnancy, prematurity, gestational diabetes, pre-eclampsia, congenital malformations, intergeneration

## Abstract

Polycystic ovary syndrome (PCOS) is a metabolic disorder with reproductive consequences. Hence, the synergy of the dual maternal challenges of difficulties with conception, set on a background of metabolic disorder and inflammation, understandably leads to increased obstetric risk for the woman. Furthermore, she is more likely than her peers to require assistance with conception, either through induction of ovulation with the attendant risk of a multiple gestation, or in vitro fertilization (IVF) with its recognized increased obstetric risk for woman and her child. The increased obstetric risk for a woman with PCOS is manifested with an increased rate of miscarriage, gestational diabetes, hypertensive disorder and premature delivery. These obstetric complications are due to impairment of placental function, systemic inflammation and metabolic disorder and are markers for the woman herself of her predisposition to cardiometabolic disorder in later life. Consequently, it is inevitable that this environment may induce changes in the fetus during pregnancy, leading to an intergenerational risk from maternal PCOS.

## 1. Introduction

Polycystic ovary syndrome (PCOS) (Table 1) [1] is a metabolic disorder with reproductive consequences [2]; consequently there exists an interplay of ovulatory dysfunction, often requiring pharmacological intervention to facilitate ovulation for conception, on a background of metabolic disorder [3,4] which is believed to predispose the woman to increased obstetric risk [5]. Hence, women with PCOS often face the double-hit of medical assistance to conceive and an increased obstetric risk upon conception and potentially, due to the increased obstetric risk, there is an increased risk to the child born [5]. Furthermore, whether it is possible to improve these intergenerational effects with intervention is unclear. This review aims to describe the potential impact of PCOS on pregnancy and the offspring and to comment on potential mechanisms and to summarize interventions that may improve any intergenerational effect of PCOS.

## 2. Polycystic Ovary Syndrome Influence on Pregnancy Outcomes

Evidence for an adverse influence of a PCOS diagnosis on pregnancy abounds, however, as PCOS consists of a spectrum of features of varying severity, the same risk does not apply to all women.

### 2.1. Miscarriage

One of the most distressing outcomes for a woman who is having difficulty in conceiving is miscarriage and unfortunately it appears that women with a diagnosis of PCOS are at an increased risk of miscarriage upon conception. Using whole population data from the very static population of Western Australia, the chance of hospitalisation for miscarriage for a woman with a diagnosis of PCOS was double that for a woman who did not have a diagnosis of PCOS [3]. This suggests that a woman who has been given the label ‘PCOS’ has an increased risk of miscarriage, which may suggest that a woman displaying more overt features of PCOS, and hence has a diagnosis, may be at greater risk. The most recent meta-analysis of the literature describes a threefold increase in the risk of miscarriage for women with PCOS, using data from three studies. This increased risk is consistent with evidence derived from both prospective and retrospective studies, although a challenge in the interpretation of the literature is that several studies do not allow for the confounding of adiposity [6]. A Chinese observational study of women undergoing in vitro fertilization (IVF) treatment did not find a difference in the chance of miscarriage when the data was controlled for body mass index (BMI) [7].

### 2.2. Obstetric Risk

The literature consistently demonstrates a significantly increased obstetric risk for a woman with PCOS and for her fetus, when compared to her peers without PCOS. Taken at face value, this would appear hardly surprising in view of the increased health risks for the woman herself outside of pregnancy, with respect to cardiovascular and metabolic disorders [3,8] as pregnancy is a substantial cardiometabolic physiological challenge for any woman. However, it would appear that, outside of pregnancy, the cardiometabolic risk to a woman with PCOS improves as she ages in relation to her peers without PCOS, eventually equating to her peers in her late reproductive years [8] so this theory may not be as plausible as it would initially appear. 

There have been many observational studies of the obstetric risk for a woman with PCOS, and these studies have undergone systematic review by Yu et al. [9]. Their review of 40 studies of 17,816 pregnant women with PCOS, 15 of which were prospective, when compared to almost 124,000 pregnant women without PCOS, included relatively young and lean women with PCOS [9]. The average age of the women included in these studies was under 33 years and similarly the average BMI was less than 29 kg/m^2^, with the majority around 22–27 kg/m^2^. The results of this review demonstrated that the pregnancy of women with PCOS was at an increased chance of a complication, with gestational diabetes almost threefold, pre-eclampsia almost threefold and a 50% increased risk of preterm delivery [9]. Interestingly, the authors did not find an association of a PCOS diagnosis with the fetus being large or small for gestational age. 

A very insightful study by Christ et al. [10] attempted to determine factors that may predict the increased obstetric risk, as PCOS is believed to be a heterogeneous condition, where anovulation may predominate for some women. However, hyperandrogenism symptoms may be the major symptom for others, hence the same obstetric risks may not apply to all. The study used a population of women from an infertility clinic with PCOS, who had undergone phenotypic assessment for features of PCOS prior to conception, and compared them to the national Dutch registry, with regard to the primary outcomes of pre-eclampsia and premature delivery. In agreement with previous studies, there was more than a twofold increase in the risk of pre-eclampsia (5% vs. 2%) for women with PCOS in comparison to the national registry data and the respective rates of premature delivery were 11% vs. 7% [10]. After regression analysis to determine the predictive markers for these adverse outcomes, the authors determined that a pre-pregnancy free androgen index was associated with subsequent pre-eclampsia (odds ratio (OR) 1.1, 95% confidence intervals (CI) 1.0–1.1) and serum fasting glucose (OR 1.4, CI 1.2–1.7) and testosterone (OR 1.5, CI 1.2–1.7) predicted preterm delivery. Furthermore, they found that a higher fasting serum insulin (OR 1.003, CI 1.001–1.005) and serum testosterone (OR 1.2, CI 1.1–1.4) were predictive of any adverse pregnancy outcome. Therefore, this study suggests that women with PCOS with hyperandrogenism have a significantly greater obstetric risk than their peers with PCOS without hyperandrogenism, providing an insight into the potential cause of the increased obstetric risk and a potential screening tool. The same group also performed another study with the same women who underwent PCOS phenotypic assessment and also demonstrated a correlation of the degree of hyperandrogenism with gestational diabetes [11].

## 3. Influence of Polycystic Ovary Syndrome on the Health of the Offspring

In line with the Barker hypothesis, whereby adverse early life influences may lead to disease later in life [12], six publications reviewed by Yu et al. as part of their meta-analysis describe an adverse association of a mother’s diagnosis of PCOS on the health of her offspring [9]. The study reports an almost threefold risk of neonatal hypoglycemia and a doubling of the rate of perinatal death, but no increased risk of fetal macrosomia, respiratory distress syndrome or fetal malformations [9]. The difficulty in the interpretation of the literature is that often the studies are heterogeneous and many have not been controlled for pre-pregnancy BMI, maternal age, multiple pregnancy and the use of IVF. Furthermore, it is also difficult to determine whether the increased risk of a premature delivery is due to the PCOS per se, or whether it is iatrogenic due to the need to expedite delivery for an obstetric indication. Such a situation could also lead to an incorrect assumption of an increase in low birth weight infants, as often the outcome was not corrected for gestational age. Such a situation may have a further flow-on effect in the analysis of the risk of congenital abnormalities in the offspring; an iatrogenic premature delivery potentially could lead to an artificial increase in the findings of undescended testicles and a patent ductus arteriosus in the infant, as these congenital abnormalities are more commonly found in premature infants, as they are a normal process of development that has not had time to be completed. We performed a large study using whole population data within Western Australia of over 2500 pregnant women with PCOS and compared them to 26,000 pregnant women without a PCOS diagnosis. The data was controlled for the use of IVF, ethnicity, maternal age, multiple gestation, maternal smoking and pre-existing co-morbidities. The main findings were that offspring of women with PCOS were twice as likely to be born prematurely, three times as likely to die in the perinatal period and twice as likely to require a postnatal hospitalisation [5]. Furthermore, offspring of women with PCOS were at an increased risk of a congenital anomaly (6.3% compared with 4.9%, OR 1.20, 95% CI 1.03–1.40) [5]. This data was additionally corrected for all obstetric risk factors, including gestational diabetes and large/small for gestational age. When the congenital abnormalities were analysed by the type of malformation, cardiovascular and urogenital malformations were more common in the offspring of women with PCOS; cardiovascular (1.5% compared with 1.0%, OR 1.37, 95% CI 1.01–1.87) and urogenital defects (2.0% compared with 1.4% OR 1.36, 95% CI 1.03–1.81). In looking beyond the neonatal period into childhood and adolescence, maternal PCOS was associated with increased hospitalisations for their offspring, including metabolic disorder, disease of the nervous system and asthma [5].

## 4. Potential Mechanisms Underlying the Increased Obstetric and Offspring Risk

It is evident that late pregnancy events will have an immediate influence on a child’s neonatal health, such as diabetes and the consequent risk of neonatal hypoglycemia and risk of neonatal unit admission. Furthermore, as described, iatrogenic intervention will have a potential influence on the immediate neonatal period. However, there are several studies that attest to the increased health related risks for the child born to a mother with PCOS well into adolescence [5] and summarized by Palomba et al. [13] which suggest that either genetic or in utero influences may be responsible for the risk of longer-term complications. The recent study by Christ et al. provides some insight into potential mechanisms of the intergenerational influence of PCOS [10]. In their study, they documented that women with hyperandrogenism prior to pregnancy were at a greater risk of pre-eclampsia and preterm delivery and that a greater serum insulin and serum testosterone were predictive of adverse pregnancy outcomes [10]. Although, in contrast, a Danish study found no differences in the obstetric outcomes when a potential influence of hyperandrogenism was sought [14].

A further study that offers a mechanistic insight into potential longer intergenerational consequences of PCOS was derived from an observational study of women with PCOS grouped according to the different definitions of PCOS [15]. The study concluded that, although the rate of maternal and neonatal complications was similar whichever definition of PCOS was used, women with PCOS by the Rotterdam criteria [1] were less likely to have impaired glucose tolerance or hyperinsulinaemia prior to conception than women with PCOS by the National Institutes of Health (NIH) criteria [16]. Consequently, the impaired glucostasis that occurs for some women with PCOS may lead to an intergenerational influence that may not be immediately evident during pregnancy, hence going undetected, which may lead to an increased risk of congenital malformations and an increased cardiometabolic risk for the offspring. 

Hyperinsulinaemia is commonly present in women with PCOS and is more prevalent when hyperandrogenic features predominate; it is exacerbated by obesity and is significantly increased when a woman with PCOS conceives [6]. Insulin promotes a prothrombotic and profibrotic environment and potentiates vascular vasoconstriction, which ultimately leads to an increase in blood pressure [6]. This environment, in concert with alterations in the uterine artery blood flow in women with PCOS [17], will inevitably influence fetal growth. Furthermore, this hyperandrogenic/hyperinsulinaemic environment has a pro-inflammatory effect [6,18], with increases in serum C-reactive protein, white blood cells, inflammatory cytokines and cell adhesion molecules [19]. This inflammatory environment, in parallel with the adverse placental changes associated with PCOS consisting of alterations in spiral arteries, placental vascular lesions and inflammation [18,20] may lead to difficulties in embryo implantation, miscarriage and an adverse pregnancy outcome, which may lead to the transgenerational influence documented. 

Women with PCOS are often hyperandrogenic in pregnancy [21], while increases in sex hormone binding protein and placental aromatase offer the fetus a degree of protection against higher levels of maternal androgens. These mechanisms to protect the fetus may be impaired in women with PCOS [22]. The consequent higher exposure to androgens may program PCOS in the offspring [23] as evidence derived from animal studies suggests treating pregnant rhesus monkeys with testosterone leads to the development of a PCOS-like phenotype in the offspring [24], providing a further potential transgenerational mechanism for the documented cardiometabolic health outcomes for the offspring noted previously.

It is also important to state that many women with PCOS require induction of ovulation for conception. If this is not performed with adequate monitoring, there is an attendant risk of multiple pregnancy and premature delivery [25] both of which are acknowledged to pose greater health risks to the offspring. Furthermore, women with PCOS are also more likely to require IVF to conceive than their peers. Children born from IVF treatment have a greater risk of congenital malformation and cardiometabolic disorder in adolescence, although the underlying mechanisms are unclear [26]. Consequently, the pregnancy risks for a woman with PCOS may be exacerbated by the dual influences of obesity and a multiple pregnancy and possibly accentuated by the need to access IVF treatment, further increasing the intergenerational risk for the offspring. 

## 5. Does Pre-Pregnancy or Obstetric Intervention Improve the Obstetric and Neonatal Outcome?

### 5.1. Lifestyle Intervention

Weight loss prior to pregnancy should be encouraged for all overweight women about to attempt to conceive, however it must be performed in a structured, multi-disciplinary manner [2] aiming for a modest weight loss of up to 10% [27] to ensure an optimal chance to conceive and to improve pregnancy outcomes. The use of bariatric surgery for weight loss for overweight women with PCOS to assist conception is not encouraged and should be considered experimental due to the significant increased perinatal risk to the infant [27]. However, despite evidence to suggest that spontaneous conception will increase, there is limited evidence to suggest that this approach will improve the health of the child born.

### 5.2. Metformin Therapy

The only insulin sensitizer that is regularly used for women with PCOS is the biguanide metformin, which also reduces gluconeogenesis, inhibits lipid synthesis and reduces gastrointestinal glucose absorption. Metformin has been used to induce ovulation for women with PCOS for many years, although it is not recommended as a first line treatment [27]. Consequently, many women conceive using metformin, therefore it is essential to ensure it is safe to take in early pregnancy. Metformin has a short half-life and readily crosses the placenta, although the extent of metformin uptake by the embryo and fetus is unclear [28]. This is particularly relevant as there is evidence to suggest that the fetal blood concentrations of metformin, using cord blood sampled at birth, may be greater than the maternal serum concentration of metformin [29].

The most recent systematic review of the literature regarding any potential effects in the offspring of the maternal use of metformin in pregnancy is generally reassuring, while acknowledging the need for more studies [30]. The meta-analysis of 10 randomized studies (778 children) of the use of metformin throughout pregnancy for women with PCOS or diabetes, in comparison to either no treatment or insulin, demonstrated that children exposed to metformin were heavier in childhood, with no other consistent clinical or metabolic findings. However there have been no long-term studies of individuals exposed to metformin in utero, as it would be expected that, if there were metabolic changes in exposed individuals, they may take many years to become evident, hence follow-up studies into adulthood are essential. 

It is important to note that, generally, in clinical practice, metformin treatment ceases at conception, or at least during the first trimester. Hence, a more important end-point is the risk of a fetal, malformation as many more women will be conceiving with the aid of metformin and ceasing its use a short time later when they realize they are pregnant. A meta-analysis of nine studies of the first trimester exposure to maternal metformin suggests that there is no increased risk of fetal malformations [31].

However, a note of caution for future investigation is raised from in vitro mouse and human fetal testicular tissue cultures, which, when exposed to metformin, have a decrease in testosterone secretion and Sertoli cell number. This, if occurring in vivo, could potentially lead to a significant adverse male reproductive transgenerational effect due to in utero metformin exposure [28,32,33].

## 6. Conclusion

PCOS is a very common reproductive and metabolic condition, consisting of a spectrum of clinical features. However, it is clear that some women with PCOS are at a greater risk of obstetric problems that may arise in pregnancy. Whether these obstetric outcomes lead to the intergenerational influence, or whether they are just a marker of the underlying metabolic disorder, is unclear. However, whatever the mechanism, it is evident that a transgenerational health influence of maternal PCOS is imparted to the offspring. Hence, it is advised that women with PCOS should endeavor to optimize their health prior to conception to reduce pregnancy-related risks, although it is unclear whether this leads to long-term improvements in the health of the offspring. 

## Figures and Tables

**Table 1 medsci-07-00049-t001:** The international evidence-based guideline for the assessment, diagnosis and management of polycystic ovary syndrome (PCOS) endorsed the Rotterdam diagnostic criteria in adults [1].

Revised 2003 Criteria (2 out of 3)
Oligo- or anovulation; Clinical and/or biochemical signs of hyperandrogenism; Polycystic ovaries and exclusion of other aetiologies (congenital adrenal hyperplasia, androgen-secreting tumours, Cushing’s syndrome).
**For the purposes of research, the guideline recommended dividing women with PCOS into different groups depending on their phenotype.** Androgen excess + ovulatory dysfunction + polycystic ovarian morphology (Phenotype A); Androgen excess + ovulatory dysfunction (Phenotype B); Androgen excess + polycystic ovarian morphology (Phenotype C); Ovulatory dysfunction + polycystic ovarian morphology (Phenotype D).

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
