# Peer review of "Generational Health Impact of PCOS on Women and Their Children"

_medsci, 2019, doi:10.3390/medsci7030049_

Reviewer 1 Report

This is an interesting manuscript with a purpose to “describe the potential impact of PCOS on pregnancy and the offspring and to comment on potential mechanisms, and to summarize interventions that may improve any intergenerational effect of PCOS.”  This is a review of the literature.

 In the abstract the authors note “These obstetric complications are not only a marker for the woman herself of cardiometabolic disorder in later life, they potentially lead to impairment of placental function, systemic inflammation and metabolic disorder in pregnancy.” Could the authors please re-write this sentence.  It sounds like the obstetric complications are leading to impairment of placental function, etc.

In the Introduction would the authors consider adding a table with criteria (at least the Rotterdam criteria and maybe NIH) used to make the diagnosis of PCOS?

Line 48:”this suggest…”  Should it be “This suggests”?

Line 75: “with gestational diabetes of almost threefold, of pre-eclampsia of almost threefold and a 50%:  Should it be “with gestational diabetes almost threefold, pre-eclampsia almost threefold and a 50%”?

In discussing the influence of PCOS on the health of offspring do the authors want to discuss the Barker Hypothesis, especially with influence on diabetes and cardiovascular disease.   

Line 112: “undescended testicles and a patent ductus arteriosus in the infant, as these congenital abnormalities”.  Are these congenital anomalies or a normal process of development that has not been completed because of prematurity? 

Line 159: “with increases on serum C-“  Should it be increases in?

Line 166: “Women with PCOS are often hyperandrogenic in pregnancy (19), while sex hormone binding”. Should it be increases in sex hormone binding protein?

Under lifestyle intervention the authors discuss weight loss aiming for a modest weight loss of up to 10%.  Do the authors have a reference for this?  Is there any role for bariatric surgery in the overweight or obese PCOS patient?

Lines 196: discusses metformin and ovulation induction.  Could the authors discuss that metformin is no longer considered a first line therapy for ovulation induction in PCOS based on the Legro et al study “Women with PCOS are often hyperandrogenic in pregnancy (19), while sex hormone binding. NEJM 2007;356:551-66.

Line 211: “changes in exposed individuals it make take many years” Should it be may take many?

Line 222: “testicular tissue cultures, which when exposed to testosterone have a decrease in testosterone:  Should it be “when exposed to metformin have a…”?

Author Response

I am very grateful to reviewer 1 for their report and I have addressed the points in turn

Reviewer report 1

This is an interesting manuscript with a purpose to “describe the potential impact of PCOS on pregnancy and the offspring and to comment on potential mechanisms, and to summarize interventions that may improve any intergenerational effect of PCOS.”  This is a review of the literature.

 In the abstract the authors note “These obstetric complications are not only a marker for the woman herself of cardiometabolic disorder in later life, they potentially lead to impairment of placental function, systemic inflammation and metabolic disorder in pregnancy.” Could the authors please re-write this sentence.  It sounds like the obstetric complications are leading to impairment of placental function, etc.

Response-thanks for the suggestion, amended to These obstetric complications are due to impairment of placental function, systemic inflammation and metabolic disorder and are a marker for the woman herself of her predisposition to cardiometabolic disorder in later life.”

In the Introduction would the authors consider adding a table with criteria (at least the Rotterdam criteria and maybe NIH) used to make the diagnosis of PCOS?

Response-thanks for the suggestion, a table added

Line 48:”this suggest…”  Should it be “This suggests”?

Response-thanks for the suggestion, corrected

Line 75: “with gestational diabetes of almost threefold, of pre-eclampsia of almost threefold and a 50%:  Should it be “with gestational diabetes almost threefold, pre-eclampsia almost threefold and a 50%”?

Response-thanks for the suggestion, amended to the suggestion

In discussing the influence of PCOS on the health of offspring do the authors want to discuss the Barker Hypothesis, especially with influence on diabetes and cardiovascular disease.   

Response-thanks for the suggestion, ref inserted and comment

Line 112: “undescended testicles and a patent ductus arteriosus in the infant, as these congenital abnormalities”.  Are these congenital anomalies or a normal process of development that has not been completed because of prematurity? 

Response-thanks for the question, they are a normal process, clarification added

Line 159: “with increases on serum C-“  Should it be increases in?

Response-thanks for the correction

Line 166: “Women with PCOS are often hyperandrogenic in pregnancy (19), while sex hormone binding”. Should it be increases in sex hormone binding protein?

Response-thanks for the correction

Under lifestyle intervention the authors discuss weight loss aiming for a modest weight loss of up to 10%.  Do the authors have a reference for this?  Is there any role for bariatric surgery in the overweight or obese PCOS patient?

Response-thanks for the suggestion, ref inserted and comment added

Lines 196: discusses metformin and ovulation induction.  Could the authors discuss that metformin is no longer considered a first line therapy for ovulation induction in PCOS based on the Legro et al study “Women with PCOS are often hyperandrogenic in pregnancy (19), while sex hormone binding. NEJM 2007;356:551-66.

Response-thanks for the suggestion, comment added – but to be clear many women are using it for weight loss, and for the PCOS guideline recommendation for it to be used with gonadotrophins for ovulation induction and also if undergoing IVF as it is a guideline recommendation to reduce risk of OHSS

Line 211: “changes in exposed individuals it make take many years” Should it be may take many?

Response-thanks for the correction

Line 222: “testicular tissue cultures, which when exposed to testosterone have a decrease in testosterone:  Should it be “when exposed to metformin have a…”?

Response-thanks for the correction

Reviewer 2 Report

This is an overall well-done and succinct review of the obstetric and fetal risks associated with PCOS in women. 

Authors did a good job acknowledging the limitations of the available data and the need for more rigorous studies. 

There is an error on page 5, line 222 where the authors stated the testicular tissue exposed to TESTOSTERONE, where I assume the exposure was metformin? 

Author Response

I am very grateful to reviewer 2 for their report and I have addressed the points in turn

Reviewer report 2

This is an overall well-done and succinct review of the obstetric and fetal risks associated with PCOS in women. 

Authors did a good job acknowledging the limitations of the available data and the need for more rigorous studies. 

Response- thanks very much for the comments

There is an error on page 5, line 222 where the authors stated the testicular tissue exposed to TESTOSTERONE, where I assume the exposure was metformin? 

Response-thanks for the correction